# Gradient-Aligned Calibration for Post-Training Quantization of Diffusion Models

**Hoang Anh Dung**   **Cuong Pham**   **Trung Le**   **Jianfei Cai**   **Thanh-Toan Do**

Department of Data Science and AI, Monash University, Australia
`{hoang.dung, cuong.pham1, trunglm, jianfei.cai, toan.do}@monash.edu`

## Abstract

Diffusion models have shown remarkable performance in image synthesis by progressively estimating a smooth transition from a Gaussian distribution of noise to a real image. Unfortunately, their practical deployment is limited by slow inference speed, high memory usage, and the computational demands of the noise estimation process. Post-training quantization (PTQ) emerges as a promising solution to accelerate sampling and reduce the memory overhead of diffusion models. Existing PTQ methods for diffusion models typically apply uniform weights to calibration samples across timesteps, which is sub-optimal since data at different timesteps may contribute differently to the diffusion process. Additionally, due to varying activation distributions and gradients across timesteps, a uniform quantization approach is sub-optimal. Each timestep requires a different gradient direction for optimal quantization, and treating them equally can lead to conflicting gradients that degrade performance. In this paper, we propose a novel PTQ method that addresses these challenges by assigning appropriate weights to calibration samples. Specifically, our approach learns to assign optimal weights to calibration samples to align the quantized model's gradients across timesteps, facilitating the quantization process. Extensive experiments on CIFAR-10, LSUN-Bedrooms, and ImageNet datasets demonstrate the superiority of our method compared to other PTQ methods for diffusion models.

## 1 Introduction

In recent years, diffusion models have become a prominent framework for high-quality image synthesis (Ho et al., 2020; Dhariwal & Nichol, 2021; Rombach et al., 2022). Despite their effectiveness, the practical deployment of diffusion models is hindered by the substantial computational cost associated with the sampling procedure, which typically involves hundreds of iterative denoising steps. Furthermore, the noise estimation networks employed in these models are often composed of complex architectures with a large number of parameters, making them resource-intensive and difficult to deploy on devices with limited computational or memory capacity.

Quantization has become a widely adopted approach for reducing the computational and memory cost of deep neural networks Dung et al. (2024); Lin et al. (2023); Jeon et al. (2023); Pham et al. (2024). More recently, this paradigm has been extended to diffusion models, where approximating network weights and activations using reduced-precision representations (Li et al. (2023); He et al. (2023b); Huang et al. (2024); Shang et al. (2023); Wang et al. (2024a)) enables substantial reductions in memory footprint and computational overhead, with only minor performance degradation. In particular, PTQ stands out as a practical method for adapting diffusion models to low-resource settings, as it allows for model compression without revisiting the training process or relying on the original dataset.

In the context of post-training quantization for diffusion models, calibration data plays a critical role in guiding the quantization process and is typically collected from various stages of the denoising trajectory. For instance, Q-Diffusion (Li et al., 2023) adopts a fixed-interval selection strategy, collecting samples uniformly across the entire set of denoising steps. On the other hand, in PTQ4DM (Shang et al., 2023), a number of timesteps are sampled from a Gaussian distribution, and the images generated at these timesteps are then used as calibration data. Building upon this, TFMQ-DM (Huang

et al., 2024) follows the same sampling strategy as Q-Diffusion and further introduces a method designed to preserve temporal feature consistency during quantization. A common assumption in existing quantization methods for diffusion models (Shang et al., 2023; Li et al., 2023; Huang et al., 2024) is that all calibration samples contribute equally to the quantization process. However, recent research on diffusion models challenges this notion by demonstrating that sample importance varies significantly across timesteps. For example, (Xie et al., 2024) shows that the loss gradient norms of samples are highly dependent on their associated timesteps, introducing a systematic bias in influence estimation. Samples corresponding to timesteps with larger gradient norms tend to exert a disproportionately higher impact on the model. Similarly, (Wang et al., 2024b) empirically categorize timesteps into acceleration, deceleration, and convergence phases based on process increments, each contributing differently to the model's learning dynamics.

On the other hand, since activations and gradients vary significantly across timesteps, calibration data from different timesteps can be interpreted as representing distinct tasks with divergent gradient dynamics. Prior works on diffusion model training have highlighted the challenge of gradient conflict, which arises when optimization directions across timesteps interfere with each other. For example, (Hang et al., 2023) frames diffusion model training as a multi-task problem, showing that optimizing the denoising objective at a specific noise level can degrade performance at others. Similarly, (Go et al., 2023) observes that negative transfer can occur due to conflicting gradients across timesteps. In the context of quantization, this challenge becomes more pronounced due to the discrete nature of the parameter space. Quantized models with binary constraints lack the flexibility to represent intermediate values, forcing parameters to take discrete values such as 0 or 1. Unlike full-precision models, which can mitigate conflicting gradient signals by adjusting parameters incrementally, quantized models cannot resolve such conflicts effectively. As a result, when gradients from different timesteps compete, the model may incur large losses in directions where no suitable quantized value exists, leading to uneven performance across timesteps. Consequently, improving performance at one timestep may inherently degrade performance at others due to representational trade-offs.

To this end, we propose a novel meta-learning–based approach that dynamically assigns importance weights to calibration samples during the quantization process. Our goal is to calibrate the quantized model using a weighted sample set that not only achieves strong validation performance but also promotes alignment between gradients from different timesteps. We formulate this as a bi-level optimization problem, learning sample weights such that the calibrated model maintains gradient consistency and improves adaptability during the quantization process. By aligning gradient directions and emphasizing samples that contribute most effectively, our method enhances gradient propagation and overall quantization quality. We validate our proposed approach on the widely used CIFAR-10 (Krizhevsky & Hinton, 2009), LSUN-Bedrooms (Yu et al., 2015) and ImageNet (Deng et al., 2009) datasets with various noise estimation network architectures under different bit-width settings. The extensive experiments demonstrate that our method outperforms the state-of-the-art PTQ methods for diffusion models. The contributions of this work can be summarized as follows:

- We are the first to identify the issue of gradient conflict during post-training quantization of diffusion models, where calibration samples from different timesteps may induce inconsistent optimization directions.
- We introduce the first PTQ framework for diffusion models that leverages gradient alignment to learn sample-wise importance weights for calibration data. By emphasizing samples with coherent gradient directions across timesteps, our method enhances quantization effectiveness.
- Extensive experiments on CIFAR-10, LSUN-Bedrooms, and ImageNet demonstrate that our approach consistently achieves superior FID scores compared to prior PTQ techniques for diffusion models.

## 2 RELATED WORKS

**Post-training quantization on diffusion Models.** Diffusion models (Ho et al., 2020; Song et al., 2021b) have emerged as a powerful generative framework, capable of producing high quality images through iterative refinement of noisy inputs. Despite their impressive results, the sheer number of inference steps required in the denoising trajectory poses a substantial bottleneck for real-world deployment. While acceleration techniques (Lu et al., 2022; Song et al., 2021a; Zhao et al., 2023)

have been introduced to reduce inference time, these methods often remain resource-intensive due to the size and complexity of the underlying noise estimation networks. To alleviate these overhead, model compression techniques, particularly model quantization (Li et al., 2023; He et al., 2023b; Wang et al., 2024a; Huang et al., 2024; Shang et al., 2023; He et al., 2023a), offer a promising solution for diffusion models, by minimizing both computational and memory footprints. Among these, the post-training quantization (PTQ)(Li et al., 2023; Huang et al., 2024) has gained much attention as a practical approach that does not require full model retraining.

**Data optimization for diffusion model quantization.** PTQ methods for diffusion models typically rely on generated calibration data and effective quantization strategies. Recent efforts in this direction have primarily focused on sampling strategies for calibration, aiming to select optimal calibration data for the quantization process. For instance, APQ-DM (Wang et al., 2024a) adopts a principled time-step selection strategy rooted in structural risk minimization to guide the generation of calibration inputs. On the other hand, PTQ4DM(Shang et al., 2023) demonstrates that calibrating quantized models with samples generated from the denoising process leads to superior results compared to using samples from the forward process. Building on this idea, Q-Diffusion (Li et al., 2023) samples intermediate results at fixed intervals and introduces a novel shortcut-aware quantization technique to improve the quantized model's performance across different benchmarks.

While existing PTQ methods typically assign equal weight to all calibration samples, this overlooks important characteristics of diffusion models. Prior studies (Nichol & Dhariwal, 2021; Zhang et al., 2022) have shown that different timesteps contribute unequally to the generative process. For instance, later timesteps tend to capture higher-level semantic structures, while earlier ones focus more on denoising low-level details. Treating all timesteps uniformly may therefore dilute the influence of more impactful samples, leading to suboptimal quantization.

Moreover, samples from different timesteps follow distinct distributions and can be seen as separate subtasks with divergent learning dynamics. Uniformly optimizing over the entire training set may induce conflicting gradient signals across timesteps, resulting in performance trade-offs where improvements in certain timesteps degrade others. To address these challenges, we introduce a meta-learning–based framework that learns sample-wise importance weights, promoting calibration samples that yield coherent and stable gradient directions across timesteps. This improves the overall quantization quality by guiding optimization in a more coherent direction.

## 3 PRELIMINARY ANALYSIS

To investigate the gradient dynamics induced by calibration samples from different timesteps during the quantization process, we analyze how gradients of the quantized model vary across timesteps. Specifically, we compute gradient vectors of the quantization loss evaluated on calibration samples drawn from different timesteps with respect to model parameters, using 256 samples per timestep for CIFAR10 dataset. We then measure the pairwise cosine distance between these gradient vectors to construct a gradient dissimilarity matrix. As shown in the heatmap in Figure 1a, the cosine distance between gradients varies across timesteps. In particular, while gradients from earlier timesteps exhibit higher consistency, those from later stages of the denoising process tend to diverge more noticeably. This observation indicates that calibration data from different timesteps induce distinct gradient signals. Ignoring these variations during quantization may result in gradient misalignment, which can hinder effective optimization and reduce generalization across timesteps, causing the model to perform well to certain timesteps while underperforming on others.

Additionally, we visualize the loss of the quantized model after calibrated, using timestep-specific calibration subsets, each corresponding to a specific timestep. The results, shown in Figure 1b, reveal significant variation in loss across timesteps, indicating that the quantized model struggles to generalize across the full diffusion process.

## 4 PROPOSED METHOD

### 4.1 PROBLEM DEFINITION

Before the quantization process starts, we construct the calibration set following the procedure outlined in Q-Diffusion (Li et al., 2023), by selecting generated samples at fixed intervals across the

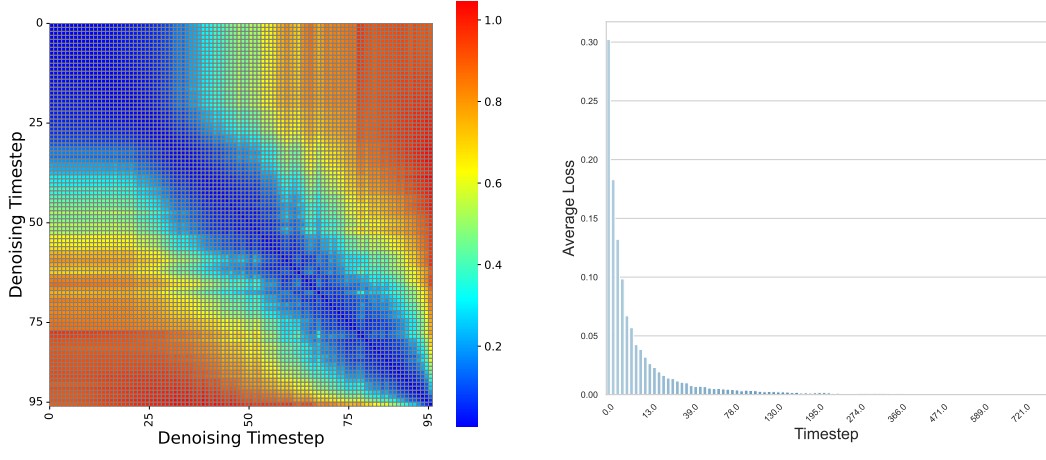

(a) Gradient dissimilarity across timesteps.                (b) Timestep-wise loss of the quantized model

Figure 1: Timestep-wise behavior in quantized diffusion models. (a) Gradient dissimilarity matrix constructed by computing pairwise cosine distance between gradient vectors of the quantization loss, with respect to model parameters, for calibration samples drawn from different timesteps. Higher value indicates higher divergent between timesteps. (b) Quantization loss evaluated separately for calibration samples grouped by timestep, highlighting the uneven performance of the quantized model across different timesteps.

denoising timesteps. Each calibration sample in the training set $X^{(T)} = \bigcup_{t=1}^{\mathbf{T}} X_t^{(T)}$ is represented as $(x_i^{(T)}, t_i^{(T)}) \in X_{t_i}^{(T)}$, where $x_i^{(T)}$ is a generated sample at the $t_i$ timestep. A validation set $X^{(V)} = \bigcup_{t=1}^{\mathbf{T}} X_t^{(V)}$ is constructed by generating an equal number of synthetic samples $X_t^{(V)}$ from each timestep $t$. In our method, each calibration sample $(x_i^{(T)}, t_i^{(T)})$ is assigned a learnable weight $\omega_i$, which reflects its influence on the quantized model's performance. The complete set of weights for all training samples is denoted by $\omega = \{\omega_i\}_{i=1}^{|X^{(T)}|}$. Our goal is to dynamically optimize training sample weights, such that the resulting quantized model $\theta_Q^*$ obtained after quantized using the weighted samples, achieves strong performance on the validation set. Given the full-precision model $\theta_{\mathrm{FP}}$ and the initial quantized model $\theta_Q$, the optimization objective for $\omega$ is defined as follows:

$$\omega = \arg\min_{\omega} \mathcal{L}_{VAL}(\theta_Q^*(\omega), \theta_{FP}, X^{(V)})$$

$$\text{s.t: } \theta_Q^*(\omega) = \theta_Q - \eta \sum_{i=1}^{|X^{(T)}|} \omega_i \frac{\partial \mathcal{L}_{MSE}(\theta_Q, \theta_{FP}, x_i^{(T)})}{\partial \theta_Q}, \tag{1}$$

where $|.|$ signifies the cardinality of a given set; $\mathcal{L}_{MSE}(.)$ denotes the quantization loss (MSE loss) that matches the outputs of the full-precision model $\theta_{FP}$ and the quantized model $\theta_Q$; $\mathcal{L}_{VAL}(.)$ denotes our optimization objective, to help the model achieves strong performance on the validation set. $\theta_Q^*(\omega)$ denotes the final quantized model after calibrated over the training set; $\eta$ denotes the learning rate.

## 4.2 CALIBRATION DATA OPTIMIZATION

Beside model performance on the validation set, since all timesteps in diffusion model shares the same quantized model weight $\theta_Q$, for a consistent quantization process, we aim to align the gradients of $\theta_Q^*$ across different timestep-specific validation set , promoting smoother optimization across subsequent quantization stages. In practice, we would divide timesteps into different groups, as adjacent timesteps often exhibit similar gradient behavior, as illustrated in Figure 1a. However, for simplicity, we assume each group consists of a single timestep and use $\mathbf{T}$ to denote both the number of timesteps and the number of groups throughout the algorithmic description.

Our validation loss $\mathcal{L}_{VAL}(.)$ is defined as:

$$\mathcal{L}_{VAL}(\theta_Q^*, \theta_{FP}, X^{(V)}) = \mathcal{L}_{GM}(\theta_Q^*, X^{(V)}) + \mathcal{L}_{MSE}(\theta_Q^*, \theta_{FP}, X^{(V)}), \tag{2}$$

where the gradient matching loss $\mathcal{L}_{GM}$ for gradients w.r.t the model weights is defined as:

$$\mathcal{L}_{GM}(\theta_Q^*, X^{(V)}) = -\frac{2}{\mathbf{T} * (\mathbf{T} - 1)} \sum_{t \neq k} \mathcal{G}_{\theta_Q^*, t} \mathcal{G}_{\theta_Q^*, k}$$

$$\text{s.t: } \mathcal{G}_{\theta_Q^*, t} = \frac{\partial \mathcal{L}_{MSE}(\theta_Q^*, \theta_{FP}, X_t^{(V)})}{\partial \theta_Q^*} \tag{3}$$

**Regarding the loss $\mathcal{L}_{MSE}$ in Eq. (1).** The reconstruction loss $\mathcal{L}_{MSE}$, commonly employed in prior quantization methods, is defined as follows:

$$\mathcal{L}_{MSE}(\theta_Q, \theta_{FP}, X^{(T)}) = \frac{1}{|X^{(T)}|} \sum_{i=1}^{|X^{(T)}|} \mathcal{L}_{MSE}(\theta_Q, \theta_{FP}, x_i)$$

$$= \frac{1}{|X^{(T)}|} \sum_{i=1}^{|X^{(T)}|} \|\mathtt{f}(\theta_{FP}, x_i) - \mathtt{f}(\theta_Q, x_i)\|^2, \tag{4}$$

where $\mathtt{f}(\theta_{FP}, x_i)$ and $\mathtt{f}(\theta_Q, x_i)$ respectively denote the outputs of the full-precision and quantized models given the input sample $x_i$.

Directly optimizing Eq. (1) is challenging due to the involvement of the third-order term in the gradient of $\mathcal{L}_{GM}(.)$ with respect to the sample weights $\omega$. To address this, we propose a more efficient algorithm in Algorithm 2, and prove that the proxy objective optimized by this algorithm serves as a faithful surrogate for the original loss in Eq.( 1). Theorem 4.1 formalizes this relationship, demonstrating that optimization via Algorithm 2 induces the minimization of the original objective in Eq. (1).

**Theorem 4.1.** *The optimization in Algorithm 2 implicitly lead to the minimization of the target objective $\mathcal{L}_{VAL}(\cdot)$ in Eq. (1).*

In order to prove our main result, we present two lemmas that will be instrumental in the proof of the theorem.

**Lemma 4.2.** *Let us denotes $\mathcal{G}_{\omega,t} = \frac{\partial \mathcal{L}_{MSE}(\theta_Q^*, \theta_{FP}, X_t^{(V)})}{\partial \omega}$. The second gradient matching loss $\mathcal{L}_{GM}^{(2)}(.)$ for gradients w.r.t the sample weight $\omega$ is defined as:*

$$\mathcal{L}_{GM}^{(2)}(\theta_Q^*, X^{(V)}) = -\frac{2}{T * (T - 1)} \sum_{t \neq k} \mathcal{G}_{\omega,t} \mathcal{G}_{\omega,k}, \tag{5}$$

*The minimization of $\mathcal{L}_{GM}^{(2)}(.)$ will implicitly lead to the minimization of $\mathcal{L}_{GM}(.)$, in the sense that a minimizer of $\mathcal{L}_{GM}^{(2)}$ corresponds to a minimizer of the target loss $\mathcal{L}_{GM}$.*

We begin by leveraging Lemma 4.2 to show that minimizing the surrogate gradient matching loss $\mathcal{L}_{GM}^{(2)}(.)$ in Eq. (15 )implies the minimization of the original gradient matching loss $\mathcal{L}_{GM}(.)$ in Eq. (3). Therefore, minimizing the surrogate validation objective $\mathcal{L}_{VAL}^{(2)}(.)$ in Eq. (8) leads to the minimization of the true validation loss $\mathcal{L}_{VAL}(.)$ in Eq. (2). To establish Theorem 4.1, it thus suffices to show that Algorithm 2 minimizes $\mathcal{L}_{VAL}^{(2)}(.)$, which is stated by Lemma 4.3:

**Lemma 4.3.** *Let us define a second validation loss:*

$$\mathcal{L}_{VAL}^{(2)}(\theta_Q^*, \theta_{FP}, X^{(V)}) = \mathcal{L}_{GM}^{(2)}(\theta_Q^*, X^{(V)}) + \mathcal{L}_{MSE}(\theta_Q^*, \theta_{FP}, X^{(V)}), \tag{6}$$

*The Algorithm 2 will minimize $\mathcal{L}_{VAL}^{(2)}(\theta_Q^*, \theta_{FP}, X^{(V)})$ in Eq. (8).*

Therefore, combining Lemma 4.3 and Lemma 4.2, we conclude that Theorem 4.1 holds. Please see the Supplementary for the proof of our Lemmas 4.3 and 4.2.

Based on Theorem 4.1, we can optimize the sample weights in Eq. (1) implicitly using the Algorithm 2 in the Appendix.

## 4.3 OVERALL OPTIMIZATION FRAMEWORK

At the beginning of training, a synthetic dataset is constructed by sampling from the full-precision diffusion model across multiple timesteps. This set is then divided into a timestep-balanced validation set and a training set. The training sample weight $\omega$ are initialized uniformly as

$$\omega_i = \frac{\exp(s_i/\tau)}{\sum_j \exp(s_j/\tau)}, \tag{7}$$

where we initialize $s_i = \frac{1}{32} \quad \forall \quad 0 \leq i < |X^{(T)}|$, $\tau$ denotes the temperature hyper-parameter. During the training process, model calibration is performed in a block-wise fashion, with sample weights updated at each transition to a new block. A summary of the proposed model weight calibration procedure is provided in Algorithm 1.

---

**Algorithm 1** Diffusion Model Quantization with Sample Weights

1: **Input:** Full-precision model $\theta_{FP}$; number of layers $L$; number of timesteps $\mathbf{T}$; the training set $X^{(T)}$; the validation set $X^{(V)}$
2: Initialize sample weights $\omega$
3: Initialize quantized model $\theta_Q \leftarrow \theta_{FP}$
4: **for** $\ell = 1$ to $L$ **do**
5:     Use Algorithm 2 to update $\omega$
6:     Update quantized model $\theta_Q$ by calibrating layer $\ell$ with training set $X^{(T)}$ and the updated sample weights $\omega$.
7: **end for**
8: **Return:** Quantized model $\theta_Q$

---

## 5 EXPERIMENTS

### 5.1 EXPERIMENTAL SETUP

**Models and datasets.** To assess the effectiveness of our approach, we conduct experiments on popular diffusion architectures. Specifically, we provide performance on DDPM (Ho et al., 2020), for unconditional generation, and LDM (Rombach et al., 2022), which utilizes latent space and supports both unconditional and class-conditional generation tasks.

Our evaluation spans multiple standard datasets, including CIFAR-10 at a resolution of $32 \times 32$(Krizhevsky et al., 2010), LSUN-Bedrooms at $256 \times 256$(Yu et al., 2015), and ImageNet at $256 \times 256$ (Deng et al., 2009).

**Implementation details.** Our approach aligns with the current state-of-the-art techniques in post-training quantization (PTQ) applied to diffusion models (Shang et al., 2023; Huang et al., 2024), targeting both model weights and activations. In practical scenarios, post-training quantization (PTQ) for diffusion models typically addresses weight and activation quantization as separate processes. For activations, TFMQ-DM (Huang et al., 2024) has shown that adopting sophisticated quantization techniques tends to introduce high computational cost while yielding only limited performance gains. As a result, we employ the lightweight activation quantization method from TFMQ-DM, which estimates activation ranges using an exponential moving average (EMA)(Jacob et al., 2018) over mini-batches. On the other hand, we quantize the model weights using the AdaRound algorithm (Nagel et al., 2020), in conjunction with block-wise reconstruction (Li et al., 2021), to efficiently quantize the noise-estimation network. We adopt AdaRound as it is the standard weight-quantization scheme used in prior diffusion PTQ methods (Huang et al., 2024; He et al., 2023b), ensuring a fair and consistent comparison. We begin by generating the calibration data through inference on the pretrained full-precision diffusion model, as outlined in Q-Diffusion (Li et al., 2023). To ensure consistency with prior post-training quantization efforts (Shang et al., 2023; Li et al., 2023; Huang et al., 2024), we adopt the LAPQ method (Nahshan et al., 2021) to initialize the quantized model parameters $\theta_Q$ using weights from the original full-precision network. Additionally, we integrate the temporal feature preservation strategy proposed in TFMQ-DM (Huang et al., 2024) to better maintain generative quality. Regarding the validation set, we use a small subset of the generated data

as the validation set. Since our algorithm aligns gradients across timesteps and adjacent timesteps tend to exhibit similar behavior, we partition the validation data into 5 groups, each corresponding to a consecutive range of timesteps. Each group is treated as a separate task in our algorithm. We optimize the sample weights using the Adam optimizer with a learning rate of $5 \times 10^{-6}$ for 1500 iterations per update. For each quantization block, we apply 20000 optimization iterations, in line with existing approaches (Shang et al., 2023; Li et al., 2023; Huang et al., 2024). Optimization of the sample weights $\omega$ is carried out using the Adam optimizer (Kingma & Ba, 2015), with a fixed learning rate of $4 \times 10^{-5}$. We employ the *higher* library[1] to enable gradient-based meta-optimization.

**Evaluation metrics.** To ensure alignment with established benchmarks (Shang et al., 2023; Li et al., 2023; Huang et al., 2024), we assess the generative quality of diffusion models using both the Fréchet Inception Distance (FID)(Heusel et al., 2017) and spatial Fréchet Inception Distance (sFID)(Salimans et al., 2016). These metrics provide complementary insights into visual fidelity and spatial coherence. Specifically, FID evaluates the discrepancy between real and generated image distributions by comparing their high-level Inception features, while sFID focuses on mid-level features to better capture localized structural patterns in the images. Following common practice, we generate and evaluate 50,000 synthetic samples to compute these scores, for a fair comparison with prior quantization studies.

Table 1: Quantization results for unconditional image generation with DDIM on CIFAR-10 $32 \times 32$.

| Methods | CIFAR-10 $32 \times 32$ | | | | | |
|---|---|---|---|---|---|---|
| | W/A | FID↓ | sFID↓ | W/A | FID↓ | sFID↓ |
| PTQ4DM (Shang et al., 2023) | | 5.65 | - | | 5.14 | - |
| Q-Diffusion (Li et al., 2023) | 4/32 | 5.08 | 4.98 | 4/8 | 4.98 | 5.68 |
| TFMQ-DM (Huang et al., 2024) | | 4.73 | - | | 4.78 | - |
| Ours | | **4.28** | **4.56** | | **4.32** | **4.61** |

Table 2: Quantization results for image generation with LDM-4 on LSUN-Bedrooms and ImageNet at resolution $256 \times 256$. We report FID, sFID, Precision, and Recall for each dataset.

| Methods | Bits (W/A) | LSUN-Bedrooms | | | | ImageNet | | | |
|---|---|---|---|---|---|---|---|---|---|
| | | FID↓ | sFID↓ | Precision↑ | Recall↑ | FID↓ | sFID↓ | Precision↑ | Recall↑ |
| Full Prec. | 32/32 | 2.98 | 7.09 | – | – | 10.91 | 7.67 | – | – |
| PTQ4DM (Shang et al., 2023) | 4/32 | 4.83 | 7.94 | – | – | – | – | – | – |
| Q-Diffusion (Li et al., 2023) | 4/32 | 4.20 | 7.66 | – | – | 11.87 | 8.76 | – | – |
| PTQD (He et al., 2023b) | 4/32 | 4.42 | 7.88 | – | – | 11.65 | 9.06 | – | – |
| TFMQ-DM (Huang et al., 2024) | 4/32 | 3.60 | 7.61 | 65.92 | 44.88 | 10.50 | 7.98 | 92.91 | 30.24 |
| Ours | 4/32 | **3.14** | **7.22** | **66.11** | **45.50** | **10.17** | **7.40** | **93.02** | **30.97** |
| PTQ4DM (Shang et al., 2023) | 4/8 | 20.72 | 54.30 | – | – | – | – | – | – |
| Q-Diffusion (Li et al., 2023) | 4/8 | 6.40 | 17.93 | – | – | 10.68 | 14.85 | – | – |
| PTQD (He et al., 2023b) | 4/8 | 5.94 | 15.16 | – | – | 10.40 | 12.63 | – | – |
| TFMQ-DM (Huang et al., 2024) | 4/8 | 3.68 | 7.65 | 65.89 | 44.99 | 10.29 | **7.35** | 92.53 | **30.98** |
| Ours | 4/8 | **3.26** | **7.40** | **66.05** | **45.20** | **9.96** | 7.55 | **92.75** | 30.71 |

## 5.2 COMPARISON WITH THE STATE-OF-THE-ART QUANTIZATION METHODS FOR DIFFUSION MODELS

To validate the effectiveness of our method, we benchmark it against the current state-of-the-art post-training quantization techniques developed for diffusion models, such as TFMQ-DM (Huang et al., 2024), PTQD (He et al., 2023b)PTQ4DM (Shang et al., 2023) and Q-Diffusion (Li et al., 2023). For consistency and comparability, the baseline results are directly adopted from the TFMQ-DM study. Our evaluation spans a diverse set of datasets: CIFAR-10 ($32 \times 32$), LSUN-Bedrooms ($256 \times 256$) for unconditional generation tasks, as well as ImageNet ($256 \times 256$) for class-conditional image synthesis. All experiments adhere to the evaluation protocols established in prior work (Huang et al., 2024).

---

[1]https://github.com/facebookresearch/higher

**Unconditional image generation.** To evaluate the effectiveness of our method under extreme low-bit quantization, we conduct comprehensive experiments on both low-resolution and high-resolution generative models. Specifically, we test DDPM on CIFAR-10 at $32 \times 32$ resolution, and LDM-4 on two high-resolution datasets: LSUN-Bedrooms at $256 \times 256$. For consistency and fair comparison, we employ the DDIM sampler (Song et al., 2021a) with 100 steps for CIFAR-10 and 200 steps for the two high-resolution datasets.

Across all three benchmarks, our method consistently outperforms existing state-of-the-art techniques, particularly in low-bit regimes. As reported in Table 2, we achieve state-of-the-art FID scores in all quantization settings. Specifically, on CIFAR-10, our approach surpasses the compared method method (TFMQ-DM) with FID improvements of $0.45$ and $0.46$ in W4A32 and W4A8 settings, respectively. Similar improvements are observed on LSUN-Bedrooms, with FID gains of $0.46$ (W4A32) and $0.42$ (W4A8). In summary, our approach demonstrates consistent and meaningful improvements over prior methods.

**Class-conditional image generation.** We further evaluate our method on the high-resolution ImageNet dataset using LDM-4 and the DDIM sampler (Song et al., 2021a) with 20 denoising steps. To ensure fair comparison, baseline results are adopted directly from the TFMQ-DM paper (Huang et al., 2024). As summarized in Table 2, our approach consistently surpasses existing methods across all quantization configurations.

In particular, the performance gain in the W4A32 setting is notable: our method reduces FID by $0.33$ and sFID by $0.58$ compared to TFMQ-DM, one of the current state-of-the-art PTQ methods for diffusion model. These results highlight our method's ability to preserve generation quality under aggressive quantization, even on large-scale and visually complex datasets such as ImageNet.

## 5.3 ABLATION STUDIES

**Ablation studies for the temperature parameter $\tau$.** Please refer to the Appendix for more ablation studies of our method. We vary the value of $\tau$ from 0.2 to 2 and evaluate the model's performance on the CIFAR-10 dataset under the W4A32 setting, as shown in Table 3b. As shown, the performance will degrade if we set **T** to smaller values.

Table 3: Ablation studies on CIFAR-10 dataset.

(a) Effect of validation set size.

| $\tau$ | 2% | 5% | 10% | 20% |
|---|---|---|---|---|
| FID↓ | 4.55 | **4.32** | 4.59 | 4.75 |
| sFID↓ | 4.71 | 4.61 | **4.38** | 4.51 |

(b) Effect of temperature parameter $T$.

| $\tau$ | 0.2 | 0.5 | 1 | 2 |
|---|---|---|---|---|
| FID↓ | 4.85 | 4.55 | **4.28** | 4.32 |
| sFID↓ | 4.74 | 4.67 | **4.56** | 4.61 |

**Ablation studies for the number of timesteps.** To evaluate the effectiveness of our method in the extreme case of very few timesteps, we test it on the ImageNet dataset (4/32 setting) using the DDIM sampler with 5, 10, and 20 timesteps. As shown in Table 4, our method remains highly effective, outperforms the baseline TFMQ-DM under these challenging conditions.

Table 4: Ablation studies for the number of inference timesteps. The results demonstrate that our method remains effective when the number of timesteps is small.

| Methods | Timestep | FID ↓ | sFID ↓ |
|---|---|---|---|
| TFMQ-DM (Huang et al., 2024) | 20 | 10.50 | 7.98 |
| Ours | 20 | **10.17** | **7.40** |
| TFMQ-DM (Huang et al., 2024) | 10 | 9.01 | 12.75 |
| Ours | 10 | **8.73** | **11.26** |
| TFMQ-DM (Huang et al., 2024) | 5 | 19.10 | 38.69 |
| Ours | 5 | **18.22** | **35.05** |

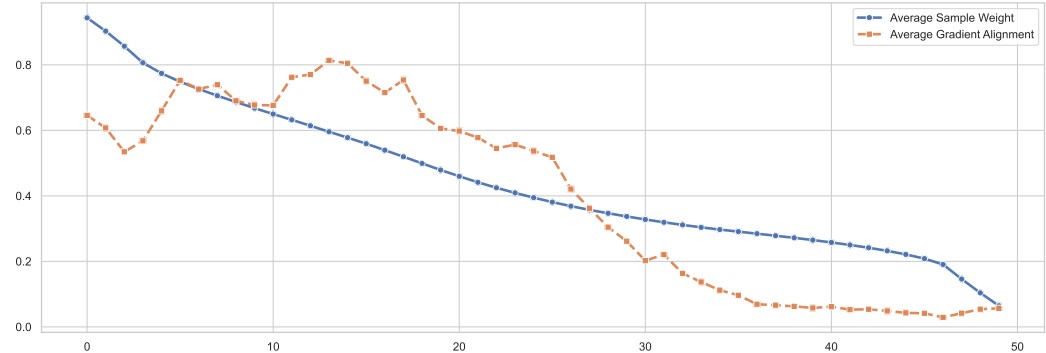

Figure 2: Visualization of the correlation between optimized sample weights and gradient alignments. All samples are sorted in descending order ot their sample weights, and divided uniformly into 50 groups. The blue line represents the average sample weight per group, while the red line indicates the average gradient alignment between samples in each group and the validation set. This demonstrates the positive correlation between gradient alignment and sample weight.

**Ablation studies for the validation set size.** In practice, we only use a small part of the training set as the validation set ( 5%), therefore the total number of images that our methods use are equal to that of the baseline method (TFMQ-DM). To assess sensitivity, we evaluate the performance using various validation sizes below in Table 3a. We observe that the method performs reliably across all sizes, with 5% achieving the best FID and competitive sFID. Notably, increasing the validation size beyond this point does not consistently improve performance, possibly due to increased sample diversity making it harder to optimize the reweighting under fixed calibration cost.

**Visualization of sample weight.** We visualize the distribution of sample weights alongside their corresponding average gradient similarity scores across groups in the validation set. For each training sample, the gradient similarity score is computed by assessing the alignment of its gradient with those of each task, and then averaging the similarity scores across tasks. As illustrated in Figure 2, our method assigns higher weights to samples exhibiting stronger gradient alignment, which facilitates more consistent optimization across the diffusion process by prioritizing samples that reduce gradient conflicts and improve the overall convergence across timesteps

**The comparison of the computation cost and hardware efficiency.** Although our method introduces additional computational overhead during training process, it remains competitively efficient. On LSUN-Bedrooms ($256 \times 256$) under the W4A8 setting, TFMQ-DM (Huang et al., 2024) and Q-Diffusion (Li et al., 2023) requires 2.32 and 5.29 GPU hours for training cost, respectively. In comparison, our approach takes around 3.5 GPU hours, representing a moderate increase by 1 hour over TFMQ-DM, but still more efficient than Q-Diffusion. Crucially, this modest increase in training cost yields consistently superior FID results across all evaluated settings, demonstrating an effective trade-off between performance and computational burden. It's also important to note that the added complexity is confined to the training stage. During inference, our method shares the same model structure and quantization format as TFMQ-DM, leading to identical hardware efficiency and latency at test time.

## 6 CONCLUSION

In this work, we address the overlooked issue of gradient conflict during post-training quantization (PTQ) of diffusion models, which arises from treating calibration samples across timesteps as equally important. We propose a meta-weighting framework that dynamically learns sample-wise importance by promoting gradient alignment across timesteps. This approach enables more effective calibration under quantization constraints. Extensive experiments on CIFAR-10, LSUN-Bedrooms, and ImageNet demonstrate consistent improvements over existing PTQ methods, highlighting the significance of timestep-aware sample weighting.

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

# A  APPENDIX

## A.1  THE USE OF LARGE LANGUAGE MODELS

We used a large language model (ChatGPT) to help with editing this paper. It was only used for simple tasks such as fixing typos, rephrasing sentences for clarity, and improving word choice. All ideas, experiments, and analyses were done by the authors, and the use of LLMs does not affect the reproducibility of our work.

## A.2  THEORETICAL PROOFS

**Lemma A.1** (Restated from Lemma 4.3). *Let us define a second validation loss:*

$$\mathcal{L}^{(2)}_{VAL}(\theta^*_Q, \theta_{FP}, X^{(V)}) = \mathcal{L}^{(2)}_{GM}(\theta^*_Q, X^{(V)}) + \mathcal{L}_{MSE}(\theta^*_Q, \theta_{FP}, X^{(V)}), \tag{8}$$

*The Algorithm 1 will minimize $\mathcal{L}^{(2)}_{VAL}(\theta^*_Q, \theta_{FP}, X^{(V)})$ in Eq. (8).*

*Proof of Lemma 1.1.* Let us denote $\mathcal{G}^{(t)}_{\omega,i} = \frac{\partial \mathcal{L}_{MSE}(\theta^*_Q, \theta_{FP}, X^{(V)}_t)}{\partial \omega^{(i-1)}}$ as the gradient w.r.t $\omega^{(i)}$ when evaluated on the validation set at the t-th iteration, $\omega^{(i-1)}$ denotes the samples weight after the (i-1)-th iteration and before the i-th iteration optimized with Algorithm 1, $\omega^{(0)}$ denotes the initial sample weights. We have:

$$\omega^{(T)} - \omega^{(0)} = -\eta(\sum_{t=1}^{\mathbf{T}} \mathcal{G}^{(t)}_{\omega,t}), \tag{9}$$

Using First Order Taylor approximation around $\omega^{(0)}$ and combine with Eq. (9) we have:

$$\begin{aligned}
\mathcal{G}^{(t)}_{\omega,t} &\approx \mathcal{G}^{(t)}_{\omega,1} + \mathcal{H}_{\omega,t}(\omega^{(t-1)} - \omega^{(0)})^T \\
&= \mathcal{G}^{(t)}_{\omega,1} - \eta \mathcal{H}_{\omega,t}(\sum_{i=1}^{t-1} \mathcal{G}^{(i)}_{\omega,i})^T \\
&= \mathcal{G}^{(t)}_{\omega,1} - O(\eta),
\end{aligned} \tag{10}$$

where $\mathcal{H}_{\omega,t}$ denotes the Hessian matrix of the model loss with respect to the sample weights $\omega$, evaluated on the validation data at the t-th iteration.

Replace $\mathcal{G}^{(i)}_{\omega,i} = \mathcal{G}^{(i)}_{\omega,1} - O(\eta) \quad \forall i = 1, 2, \ldots t-1$ we have:

$$\begin{aligned}
\mathcal{G}^{(t)}_{\omega,t} &= \mathcal{G}^{(t)}_{\omega,1} - \eta \mathcal{H}_{\omega,t}(\sum_{i=1}^{t-1} \mathcal{G}^{(i)}_{\omega,i})^T \\
&\approx \mathcal{G}^{(t)}_{\omega,1} - \eta \mathcal{H}_{\omega,t}(\sum_{i=1}^{t-1} (\mathcal{G}^{(i)}_{\omega,1} - O(\eta)))^T \\
&\approx \mathcal{G}^{(t)}_{\omega,1} - \eta \mathcal{H}_{\omega,t}(\sum_{i=1}^{t-1} (\mathcal{G}^{(i)}_{\omega,1})) + O(\eta^2)
\end{aligned} \tag{11}$$

For any $i \leq j$, because at each iteration $i$ and $j$ we randomly sample validation data of a timestep, the iteration indexes are interchangeable, i.e. $\mathbb{E}[\mathcal{H}_{\omega,j}\mathcal{G}^{(i)}_{\omega,1}] = \mathbb{E}[\mathcal{H}_{\omega,i}\mathcal{G}^{(j)}_{\omega,1}]$. Therefore, we have:

$$\begin{aligned}
\mathbb{E}[\frac{\partial \mathcal{G}^{(i)}_{\omega,1}{}^T \mathcal{G}^{(j)}_{\omega,1}}{\partial \omega}] &= \mathbb{E}[\mathcal{H}_{\omega,j}\mathcal{G}^{(i)}_{\omega,1} + \mathcal{H}_{\omega,i}\mathcal{G}^{(j)}_{\omega,1}] \\
&= 2\mathbb{E}[H_{\omega,j}\mathcal{G}^{(i)}_{\omega,1}] \\
\implies \mathcal{H}_{\omega,t}(\sum_{i=1}^{t-1} (\mathcal{G}^{(i)}_{\omega,1})) &= \frac{1}{2} \sum_{i=1}^{t-1} \frac{\partial \mathcal{G}^{(i)}_{\omega,1}{}^T \mathcal{G}^{(t)}_{\omega,1}}{\partial \omega}
\end{aligned} \tag{12}$$

Combine Eq. (11) and Eq. (12) we have:

$$\mathcal{G}_{\omega,t}^{(t)} \approx \mathcal{G}_{\omega,1}^{(t)} - \eta \mathcal{H}_{\omega,t}(\sum_{i=1}^{t-1}(\mathcal{G}_{\omega,1}^{(i)}))$$

$$= \mathcal{G}_{\omega,1}^{(t)} - \frac{\eta}{2}\sum_{i=1}^{t-1}\frac{\partial \mathcal{G}_{\omega,1}^{(i)}{}^T \mathcal{G}_{\omega,1}^{(t)}}{\partial\omega} \tag{13}$$

Combine Eq. (9 )and Eq. (13) we have:

$$\omega^{(T)} - \omega^{(0)} = -\eta(\sum_{t=1}^{\mathbf{T}}\mathcal{G}_{\omega,t}^{(t)})$$

$$= -\eta(\sum_{t=1}^{\mathbf{T}}(\mathcal{G}_{\omega,1}^{(t)} - \frac{\eta}{2}\sum_{i=1}^{t-1}\frac{\partial \mathcal{G}_{\omega,1}^{(i)}{}^T \mathcal{G}_{\omega,1}^{(t)}}{\partial\omega})),$$

$$= -\eta(\sum_{t=1}^{\mathbf{T}}\mathcal{G}_{\omega,1}^{(t)} - \frac{\eta}{2}\sum_{1\le i<j\le T}\frac{\partial \mathcal{G}_{\omega,1}^{(i)}{}^T \mathcal{G}_{\omega,1}^{(j)}}{\partial\omega}),$$

$$= -\eta(\sum_{t=1}^{\mathbf{T}}\frac{\partial \mathcal{L}_{MSE}(\theta_Q^*,\theta_{FP},X_t^{(V)})}{\partial\omega} + \frac{\eta T(T-1)}{4}\frac{\partial \mathcal{L}_{GM}^{(2)}(\theta_Q^*,X^{(V)})}{\partial\omega}),$$

$$= -\eta(\frac{\partial \mathcal{L}_{MSE}(\theta_Q^*,\theta_{FP},X^{(V)})}{\partial\omega} + \frac{\eta T(T-1)}{4}\frac{\partial \mathcal{L}_{GM}^{(2)}(\theta_Q^*,X^{(V)})}{\partial\omega}), \tag{14}$$

According to Eq.( 14), the update in $\omega$ of Algorithm 1 corresponds to a gradient descent step on a composite loss function comprising $\mathcal{L}_{MSE}(.)$ and $\mathcal{L}_{GM}^{(2)}(.)$. This indicates that the optimization of Algorithm 1 effectively minimizes the combined validation loss $\mathcal{L}_{VAL}^{(2)}(\theta_Q^*,\theta_{FP},X^{(V)})$ as defined in Equation (8).

$\square$

**Lemma A.2** (Restated from Lemma 4.2). *Let us denotes* $\mathcal{G}_{\omega,t} = \frac{\partial \mathcal{L}_{MSE}(\theta_Q^*,\theta_{FP},X_t^{(V)})}{\partial\omega}$. *The second gradient matching loss* $\mathcal{L}_{GM}^{(2)}(.)$ *for gradients w.r.t the sample weight* $\omega$ *is defined as:*

$$\mathcal{L}_{GM}^{(2)}(\theta_Q^*,X^{(V)}) = -\frac{2}{T*(T-1)}\sum_{t\neq k}\mathcal{G}_{\omega,t}\mathcal{G}_{\omega,k}, \tag{15}$$

*The minimization of* $\mathcal{L}_{GM}^{(2)}(.)$ *will implicitly lead to the minimization of* $\mathcal{L}_{GM}(.)$, *in the sense that a minimizer of* $\mathcal{L}_{GM}^{(2)}$ *corresponds to a minimizer of the target loss* $\mathcal{L}_{GM}$.

*Proof of Lemma 1.2.* To demonstrate that minimizing $\mathcal{L}_{GM}^{(2)}(\cdot)$ leads to the minimization of $\mathcal{L}_{GM}(\cdot)$, we show that under sufficiently small learning rate $\eta$, the optimality of $\mathcal{L}_{GM}^{(2)}(\cdot)$ implies the optimality of $\mathcal{L}_{GM}(\cdot)$. Let $\{\mathcal{G}_{\theta_Q,t}^{(T)}\}_{t=1}^{\mathbf{T}}$ denotes the set of gradient w.r.t the intial quantized model's parameters $\theta_Q$ when evaluated on $\mathbf{T}$ timestep-specific training sets $\{X_t^{(T)}\}_{t=1}^{\mathbf{T}}$. Similarly, we define $\{\mathcal{G}_{\theta_Q^*,t}^{(T)}\}_{t=1}^{\mathbf{T}}$ as the set of gradient w.r.t the meta model's parameters $\theta_Q^*$ when evaluated on $\mathbf{T}$ timestep-specific training subsets. We aim to prove that, under sufficiently small learning rate $\eta$, for any $1 \le i < j \le \mathbf{T}$, if

$$\cos(\mathcal{G}_{\omega,i},\mathcal{G}_{\omega,j}) = 1,$$

then it follows that

$$\cos\left(\mathcal{G}_{\theta_Q^*,i},\mathcal{G}_{\theta_Q^*,j}\right) = 1.$$

We begin by expressing the gradients using the chain rule:

$$
\begin{aligned}
\mathcal{G}_{\omega,t} &= \frac{\partial \mathcal{L}_{MSE}(\theta_Q^*, \theta_{FP}, X_t^{(V)})}{\partial \omega}^T \\
&= \frac{\partial \mathcal{L}_{MSE}(\theta_Q^*, \theta_{FP}, X_t^{(V)})}{\partial \theta_Q^*}^T \frac{\partial \theta_Q^*}{\partial \omega} \\
&= \frac{\partial \mathcal{L}_{MSE}(\theta_Q^*, \theta_{FP}, X_t^{(V)})}{\partial \theta_Q^*}^T \frac{\partial \theta_Q^*}{\partial \omega} \\
&= \mathcal{G}_{\theta_Q^*,t}^T \frac{\partial (\theta_Q - \eta \sum_{i=1}^{|X^{(T)}|} \omega_i \frac{\partial \mathcal{L}_{MSE}(\theta_Q, \theta_{FP}, x_i^{(T)})}{\partial \theta_Q}) z}{\partial \omega} \\
&= -\eta \mathcal{G}_{\theta_Q^*,t}^T \left[ \frac{\mathcal{L}_{MSE}(\theta_Q, \theta_{FP}, x_1^{(T)})}{\partial \theta_Q} \quad \frac{\mathcal{L}_{MSE}(\theta_Q, \theta_{FP}, x_2^{(T)})}{\partial \theta_Q} \quad \cdots \quad \frac{\mathcal{L}_{MSE}(\theta_Q, \theta_{FP}, x_1^{(N)})}{\partial \theta_Q} \right]
\end{aligned}
\tag{16}
$$

Let us define a matrix $C$ size $T \times N$ where $C_{t,j} = cos(\mathcal{G}_{\theta_Q^*,t}, \frac{\mathcal{L}_{MSE}(\theta_Q, \theta_{FP}, x_j^{(T)})}{\partial \theta_Q})$ denoting the cosine similarity between the meta model's gradient evaluated on the validation data at timestep $t$ and the original model's gradient when evaluated on $j^{th}$ training samples.

From Eq. (16), for any $1 \le i < j \le T$, if $cos(\mathcal{G}_{\omega,i}, \mathcal{G}_{\omega,j}) = 1$, then we have:

$$
\frac{C_{i,1}}{C_{j,1}} = \frac{C_{i,2}}{C_{j,2}} = \cdots = \frac{C_{i,N}}{C_{j,N}} > 0 \qquad \forall k \in \{1, \ldots, N\} \text{ with } C_{j,k} \ne 0
\tag{17}
$$

Supposed $i = 1$, $j = 2$, then $cos(\mathcal{G}_{\omega,1}, \mathcal{G}_{\omega,2}) = 1$ and we assume that $cos\left(\mathcal{G}_{\theta_Q^*,1}, \mathcal{G}_{\theta_Q^*,2}\right) = \gamma < 1$, we will prove that this lead to contradiction. Using First-Order Taylor approximation around $\theta_Q$ we have:

$$
\begin{aligned}
\frac{\mathcal{G}_{\theta_Q^*,1}}{\left\| \mathcal{G}_{\theta_Q^*,1} \right\|} &= \frac{\mathcal{G}_{\theta_Q^*,1}^{(T)}}{\left\| \mathcal{G}_{\theta_Q^*,1}^{(T)} \right\|} \qquad (X_1^{(T)} \text{ and } X_1^{(V)} \text{ has similar distribution}) \\
&\approx \frac{1}{\left\| \mathcal{G}_{\theta_Q^*,1}^{(T)} \right\|} (\mathcal{G}_{\theta_Q,1}^{(T)} + (\theta_Q^* - \theta_Q)\mathcal{H}_1) \\
&= \frac{1}{\left\| \mathcal{G}_{\theta_Q^*,1}^{(T)} \right\|} (\mathcal{G}_{\theta_Q,1}^{(T)} - \eta \sum_{i=1}^{|X^{(T)}|} \omega_i \frac{\partial \mathcal{L}_{MSE}(\theta_Q, \theta_{FP}, x_i^{(T)})}{\partial \theta_Q} \mathcal{H}_1),
\end{aligned}
\tag{18}
$$

where $\mathcal{H}_1$ denotes the Hessian matrix w.r.t the model weight $\theta_Q$ when evaluated on the training set $X_1^{(T)}$ for the timestep 1.

Let $B = \frac{\mathcal{G}_{\theta_Q^*,1}}{\left\| \mathcal{G}_{\theta_Q^*,1} \right\|} - \frac{\mathcal{G}_{\theta_Q^*,2}}{\left\| \mathcal{G}_{\theta_Q^*,2} \right\|}$, we have

$$
\|B\| = \left\| \frac{\mathcal{G}_{\theta_Q^*,1}}{\left\| \mathcal{G}_{\theta_Q^*,1} \right\|} - \frac{\mathcal{G}_{\theta_Q^*,2}}{\left\| \mathcal{G}_{\theta_Q^*,2} \right\|} \right\| \le \left\| \frac{\mathcal{G}_{\theta_Q^*,1}}{\left\| \mathcal{G}_{\theta_Q^*,1} \right\|} \right\| + \left\| \frac{\mathcal{G}_{\theta_Q^*,2}}{\left\| \mathcal{G}_{\theta_Q^*,2} \right\|} \right\| = 2
\tag{19}
$$

Therefore, multiply both size of Eq. (18) with $B$ we have::

$$
\begin{aligned}
(\frac{\mathcal{G}_{\theta_Q^*,1}}{\left\| \mathcal{G}_{\theta_Q^*,1} \right\|})^T B &= \frac{1}{\left\| \mathcal{G}_{\theta_Q^*,1}^{(T)} \right\|} (\mathcal{G}_{\theta_Q,1}^{(T)} - \eta \sum_{i=1}^{|X^{(T)}|} \omega_i \frac{\partial \mathcal{L}_{MSE}(\theta_Q, \theta_{FP}, x_i^{(T)})}{\partial \theta_Q} \mathcal{H}_1)^T B \\
\implies \frac{1}{\left\| \mathcal{G}_{\theta_Q^*,1}^{(T)} \right\|} (\mathcal{G}_{\theta_Q,1}^{(T)})^T B &= (\frac{\mathcal{G}_{\theta_Q^*,1}}{\left\| \mathcal{G}_{\theta_Q^*,1} \right\|})^T B + \eta \frac{1}{\left\| \mathcal{G}_{\theta_Q^*,1}^{(T)} \right\|} (\sum_{i=1}^{|X^{(T)}|} \omega_i \frac{\partial \mathcal{L}_{MSE}(\theta_Q, \theta_{FP}, x_i^{(T)})}{\partial \theta_Q} \mathcal{H}_1)^T B \\
&= 1 - \gamma \quad + \eta \frac{1}{\left\| \mathcal{G}_{\theta_Q^*,1}^{(T)} \right\|} (\sum_{i=1}^{|X^{(T)}|} \omega_i \frac{\partial \mathcal{L}_{MSE}(\theta_Q, \theta_{FP}, x_i^{(T)})}{\partial \theta_Q} \mathcal{H}_1)^T B
\end{aligned}
\tag{20}
$$

Let us denote $N^{(T)} = \max_{x_i^{(T)}} \left\| \frac{\partial \mathcal{L}_{MSE}(\theta_Q, \theta_{FP}, x_i^{(T)})}{\partial \theta_Q} \right\|$. Because $\sum_i \omega_i = 1$ and $\|B\| \leq 2$ according to Eq. (19), we then have:

$$
\left| \eta \frac{1}{\left\| \mathcal{G}_{\theta_Q^*,1}^{(T)} \right\|} \left( \sum_{i=1}^{|X^{(T)}|} \omega_i \frac{\partial \mathcal{L}_{MSE}(\theta_Q, \theta_{FP}, x_i^{(T)})}{\partial \theta_Q} \mathcal{H}_1 \right)^T B \right| \leq \frac{\eta}{\left\| \mathcal{G}_{\theta_Q^*,1}^{(T)} \right\|} \|\mathcal{H}_1\|_2 \left\| \sum_{i=1}^{|X^{(T)}|} \omega_i \frac{\partial \mathcal{L}_{MSE}(\theta_Q, \theta_{FP}, x_i^{(T)})}{\partial \theta_Q} \right\| \|B\|
$$

$$
\leq 2\eta \|\mathcal{H}_1\|_2 \frac{N^{(T)}}{\left\| \mathcal{G}_{\theta_Q^*,1}^{(T)} \right\|},
$$

$$(21)$$

where $\|\mathcal{H}_1\|_2$ denotes the spectral norm of $\mathcal{H}_1$. Combining Eq. (21) and Eq. (20) we have:

$$
\frac{1}{\left\| \mathcal{G}_{\theta_Q^*,1}^{(T)} \right\|} (\mathcal{G}_{\theta_Q,1}^{(T)})^T B = \quad 1 - \gamma \quad + \eta \frac{1}{\left\| \mathcal{G}_{\theta_Q^*,1}^{(T)} \right\|} \left( \sum_{i=1}^{|X^{(T)}|} \omega_i \frac{\partial \mathcal{L}_{MSE}(\theta_Q, \theta_{FP}, x_i^{(T)})}{\partial \theta_Q} \mathcal{H}_1 \right)^T B
$$

$$
\geq \quad 1 - \gamma \quad - 2\eta \|\mathcal{H}_1\|_2 \frac{N^{(T)}}{\left\| \mathcal{G}_{\theta_Q^*,1}^{(T)} \right\|} > 0 \quad \text{for } \eta < \frac{(1-\gamma)\left\| \mathcal{G}_{\theta_Q^*,1}^{(T)} \right\|}{2\|\mathcal{H}_1\|_2 N^{(T)}}
$$

$$(22)$$

Therefore, for sufficiently small learning rate $\eta$, we have $(\mathcal{G}_{\theta_Q,1}^{(T)})^T B > 0$. This implies that there exist at least a single training sample $x_k^{(T)} \in X^{(T)}$ such that

$$
\frac{\partial \mathcal{L}_{MSE}(\theta_Q, \theta_{FP}, x_k^{(T)})}{\partial \theta_Q}^T B > 0
$$

$$
\implies \frac{\partial \mathcal{L}_{MSE}(\theta_Q, \theta_{FP}, x_k^{(T)})}{\partial \theta_Q}^T \left( \frac{\mathcal{G}_{\theta_Q^*,1}}{\left\| \mathcal{G}_{\theta_Q^*,1} \right\|} - \frac{\mathcal{G}_{\theta_Q^*,2}}{\left\| \mathcal{G}_{\theta_Q^*,2} \right\|} \right) > 0
$$

$$
\implies C_{1,k} > C_{2,k}
$$

$$(23)$$

Similarly, there exists a training sample $x_l^{(T)}$ such that $C_{2,l} > C_{1,l}$. This leads to a contradiction, because by assumption, $\frac{C_{1,k}}{C_{2,k}} = \frac{C_{1,l}}{C_{2,l}} > 0$ for all $k, l$ such that $C_{2,k} \neq 0$ and $C_{2,l} \neq 0$. If $C_{2,k} = 0$, then it must be that $C_{1,k} = 0$ as well; contradicting $C_{1,k} > C_{2,k}$. Therefore, if $\cos(\mathcal{G}_{\omega,i}, \mathcal{G}_{\omega,j}) = 1$, it follows that $\cos\left( \mathcal{G}_{\theta_Q^*,i}, \mathcal{G}_{\theta_Q^*,j} \right) = 1$ as well.

$\square$

## A.3 VISUALIZATION OF GENERATED IMAGES

We visualize sample images generated from the full-precision model, as well as from quantized models obtained using the Q-Diffusion (Li et al., 2023) method, the TFMQ (Huang et al., 2024) method, and our proposed method with the W4A32 setting, all initialized with a fixed random seed. As shown in Figure 3, our proposed method generates images that closely match those of the full-precision models, demonstrating the effectiveness of our approach.

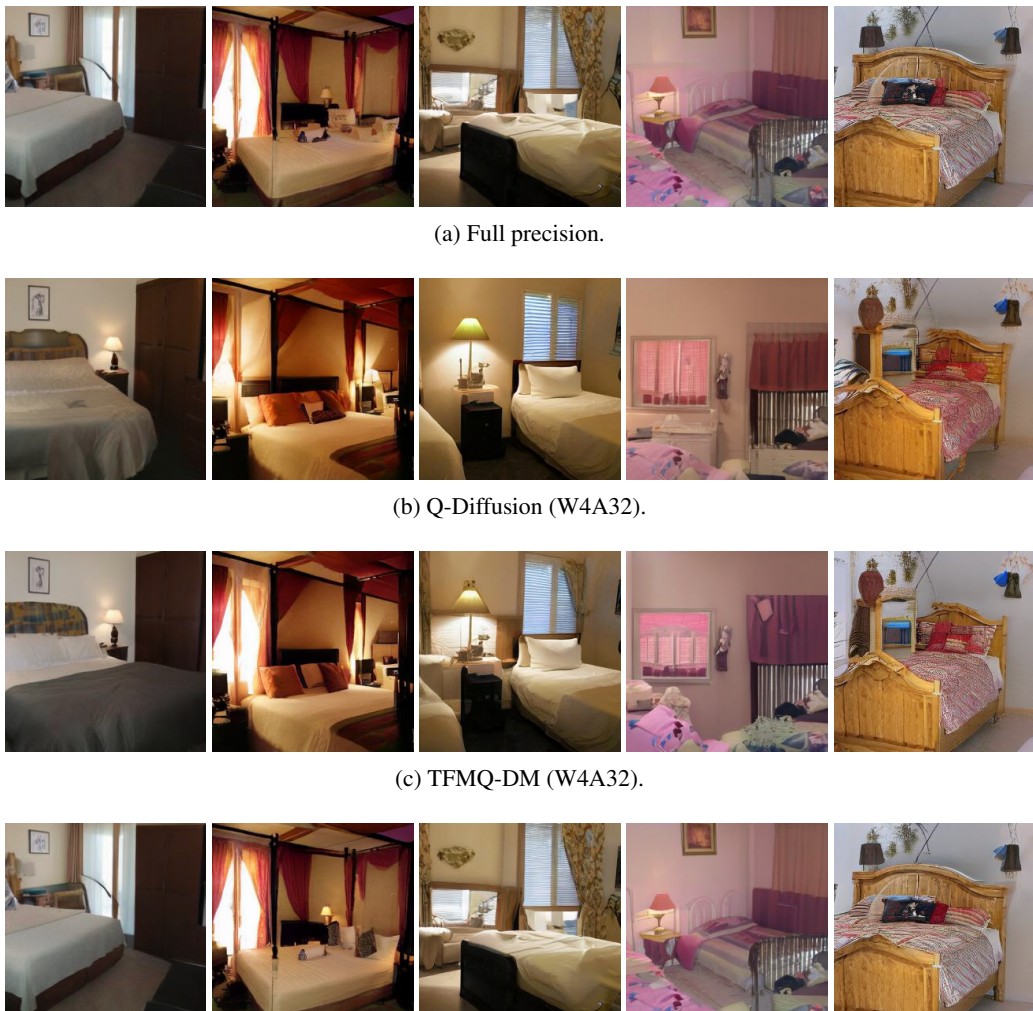

(a) Full precision.

(b) Q-Diffusion (W4A32).

(c) TFMQ-DM (W4A32).

(d) Our proposed method (W4A32).

Figure 3: Generated samples from (a) full-precision LDM-4, (b) Q-Diffusion (W4A32), (c) TFMQ-DM (W4A32), and (d) our proposed method (W4A32) on LSUN-Bedrooms $256 \times 256$ dataset with a fixed random seed.

## A.4 Final Algorithm

---

**Algorithm 2** Sample Weights Optimization for Diffusion Quantization

---

1: **Input:** Full-precision model $\theta_{FP}$; number of timesteps $\mathbf{T}$; samples per timestep $B$; learning rate for model quantization $\eta$; learning rate for sample weights $\eta_\omega$; total iterations $I$
2: Initialize quantized model $\theta_Q \leftarrow \theta_{FP}$
3: Initialize sample weights $\omega_i \leftarrow \frac{1}{B} \quad \forall i \in \{1, \ldots, \mathbf{T}\}$
4: $\omega^{(0)} = \omega$
5: **for** iteration $i = 1$ to $I$ **do**
6:     Sample timestep $t \sim \{1, \ldots, \mathbf{T}\}$               *// randomly choose a timestep*
7:     $\theta_Q^* = \theta_Q - \eta \cdot \sum_{j=1}^{B} \omega_j \cdot \nabla_{\theta_Q} \mathcal{L}_{\text{MSE}}(\theta_Q, \theta_{FP}, X^{(T)})$        *// one-step-ahead model*
8:     **for** $j = 1$ to $B$ **do**
9:         $\omega_j^{(i)} = \omega_j^{(i-1)} - \eta \cdot \nabla_{\omega_j^{(i-1)}} \mathcal{L}_{\text{MSE}}(\theta_Q^*, \theta_{FP}, X_i^{(V)})$    *// pseudo update $\omega$ with $X_i^{(V)}$*
10:     **end for**
11:     **if** $i \mod T = 0$ **then**
12:         **for** $j = 1$ to $B$ **do**
13:             $\omega_j \leftarrow \omega_j^{(0)} + \eta_\omega \cdot \frac{1}{\mathbf{T}} \left( \omega_j^{(T)} - \omega_j^{(0)} \right)$        *// final update of $\omega$*
14:         **end for**
15:         $\omega^{(0)} = \omega$
16:     **end if**
17: **end for**
18: **Return:** $\omega$

---

## A.5 Additional Results

**Ablation studies for the number of timestep groups.** In practice, we uniformly divide timesteps into 5 groups for simplicity across all datasets. To investigate the effectiveness of our method under different timestep groupings, we evaluate it on the CIFAR-10 dataset using the W4A32 setting. Table 5 presents ablation results for varying numbers of groups. We observe that increasing the number of groups does not significantly improve performance but may introduce additional computational overhead.

Table 5: Ablation studies for the number of timestep groups. The results are on the CIFAR-10 dataset with the W4A32 setting.

| Group | 1 | 2 | 5 | 10 | 20 |
|---|---|---|---|---|---|
| FID $\downarrow$ | 4.63 | 4.57 | 4.28 | 4.25 | 4.26 |
| sFID $\downarrow$ | 4.61 | 4.63 | 4.56 | 4.57 | 4.56 |

**Evaluation on extreme bit settings** To further investigate the effectiveness of our method under extreme low-bit settings, we report additional results on the ImageNet $256 \times 256$ dataset using the LDM-DM model with 3/6 and 2/4 configurations. The results are summarized in Table 6. As shown, our method achieves substantial improvements over baseline approaches, even under these highly constrained quantization regimes.

Table 6: Additional results on low-bit settings for unconditional image generation with LDM-4 on ImageNet 256×256. Results are taken from (Feng et al., 2024)

| Methods | Bits (W/A) | FID ↓ | sFID ↓ | Precision ↑ |
|---|---|---|---|---|
| Full Prec. | 32/32 | 9.36 | 8.67 | – |
| PTQD (He et al., 2023b) | | 17.98 | 57.31 | 63.13 |
| TFMQ-DM (Huang et al., 2024) | 3/6 | 15.90 | 40.63 | 67.42 |
| Ours | | **10.65** | **11.94** | **72.57** |
| PTQD (He et al., 2023b) | | 336.57 | 288.42 | 0.01 |
| TFMQ-DM (Huang et al., 2024) | 2/4 | 300.03 | 272.64 | 0.03 |
| Ours | | **226.27** | **102.83** | **0.08** |

### A.6 VISUALIZATION OF THE CHANGE IN LOSS ACROSS GROUP.

We visualize the difference in training loss between the quantized diffusion model trained with our sample-weighting strategy and one trained with uniform sample weights, using the CIFAR-10 dataset under the 4/32 quantization setting. The comparison is made across groups of timesteps sorted by ascending loss. As shown in Figure 4, our method consistently reduces the loss in groups that tend to be under-optimized when uniform weights are used. This supports our motivation to assign sample weights that mitigate gradient conflicts, preventing the model from over-optimizing certain timesteps at the expense of others.

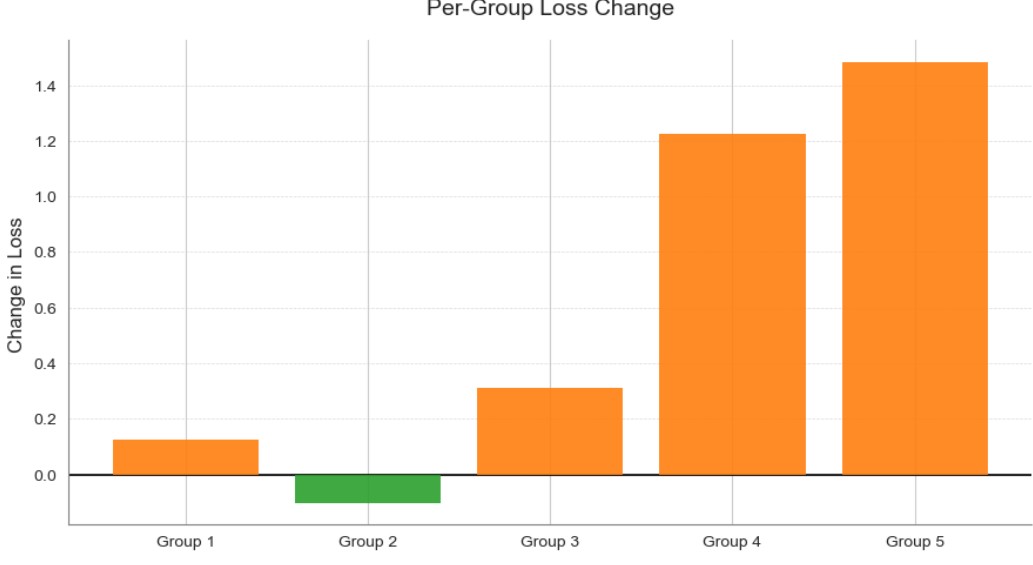

Figure 4: Visualization of loss differences across timestep groups on the CIFAR-10 calibration set (4/32 setting) for quantized models trained with and without our sample-weighting method. Data are grouped by timesteps into five categories during sample weight optimization and are shown in **ascending** order of loss. Orange bars indicate loss reduction with our method, while green bars indicate an increase. The results demonstrate that our approach effectively reduces loss for under-optimized timesteps, addressing the gradient conflict issue that leads to the neglect of certain timestep

We visualize the difference in training loss between the quantized diffusion model trained with our sample-weighting strategy and one trained with uniform sample weights, using the CIFAR-10 dataset under the 4/32 quantization setting. The comparison is made across groups of timesteps sorted by

**ascending** loss. As shown in Figure 4, our method consistently reduces the loss in groups that tend to be under-optimized when uniform weights are used. This supports our motivation to assign sample weights that mitigate gradient conflicts, preventing the model from over-optimizing certain timesteps at the expense of others.

