# OpenReview forum: "Gradient-Aligned Calibration for Post-Training Quantization of Diffusion Models"
_ICLR.cc/2026/Conference — ICLR 2026 Poster_

### Official Review · Reviewer_DoWQ · 2025-10-27

**Soundness:** 3
**Presentation:** 3
**Contribution:** 3
**Rating:** 8
**Confidence:** 4

**Summary:**

This work presents an interesting phenomenon that the quantization of diffusion models would raise grad conflict when training with different samples. With this in mind, a train-able weight is added to each sample for quantization training, to try to align the direction of grads when training quantization for different de-noising timesteps across different samples. Significant performance boosts are observed and detailed theoretical proof are offered to support the proposed motivation, as well as the methods to avoid such dis-alignment.

**Strengths:**

1. The proposed motivation is very interesting and important. It starts from the numerical angle, looking into the grad dis-alignment when training quantization for different de-noising steps, which I believe opens up an important area to explore.
2. The proposed method is well-designed. Both intuitions and theoretical proofs are provided, making it very clear to me.
3. The proposed method achieves significant performance boost on the quantization task, and theoretical analysis proves that such improvements come from solving the proposed grad misalignment problem.

**Weaknesses:**

1. Minor issue: All citations are not in the correct format, which makes reading sometimes hard. My recommendation is: the authors should check them in the next version.
2. The visualizations could be further refined to make it more impressive: While Fig. 1(a) presents the interesting grad conflict phenomenon very clearly, Fig. 2 looks not as straight-forward as Fig. 1. I recommend the authors to re-make Fig. 2 in the form of Fig. 1, to make it more impressive and more comparable.

**Questions:**

1. Can this method also be extended to diffusion model's training? Have the authors tested whether diffusion's training from scratch / fine-tuning would encounter similar phenomenon? If so, I think this can be a great extension to the manuscript. (but I don't expect the authors to add this during the short rebuttal period, and this does not negatively influence on my evaluations.)
2. While the authors ablate the results on the validation set size, I hope the authors to give some insights on the whole quantization dataset's size: when the whole dataset's size goes up, there might be more weights to learn, which might be a burden. I hope the authors can clarify on this.

**Details Of Ethics Concerns:**

I don't notice any ethics issue in this manuscript.

---

> ### Author Response · Authors · 2025-11-21
> **Response to Reviewer DoWQ**
>
> We greatly appreciate the time and effort the Reviewer dedicated to considering our paper. Here are our responses to all concerns raised by the Reviewer.
>
> ---
>
> > **W1**: All citations are not in the correct format, which makes reading sometimes hard. My recommendation is: the authors should check them in the next version.
>
> **Answer:**
> We thank the Reviewer for the valuable feedback. We have fixed the citations accordingly.
>
> ---
>
> > **W2** : The visualizations could be further refined to make it more impressive: While Fig. 1(a) presents the interesting grad conflict phenomenon very clearly, Fig. 2 looks not as straight-forward as Fig. 1. I recommend the authors to re-make Fig. 2 in the form of Fig. 1, to make it more impressive and more comparable.
>
> **Answer:**
> We thank the reviewer for the valuable suggestion. While we explored visualizing Fig. 2 in a format similar to Fig. 1, this was not directly feasible because Fig. 2 involves independent per-sample weights rather than pairwise quantities, making a heatmap-like representation inappropriate.
>
> Instead, we substantially improved Fig. 2 by:
> - sorting samples in descending order of their learned weights,
> - dividing them into 50 groups, and
> - plotting two curves: (1) the gradient alignment score of each group (average gradient cosine similarity to the validation set), and (2) the average sample weight per group.
>
> This redesigned visualization is more interpretable and clearly shows the positive correlation between gradient alignment and the learned sample weights.
>
> ---
>
> > **Q1** : Can this method also be extended to diffusion model's training? Have the authors tested whether diffusion's training from scratch / fine-tuning would encounter similar phenomenon? If so, I think this can be a great extension to the manuscript. (but I don't expect the authors to add this during the short rebuttal period, and this does not negatively influence on my evaluations.)?
>
> **Answer:**
> We appreciate the reviewer’s insightful question. Similar gradient-related issues have indeed been explored in the context of diffusion training, although not specifically for quantization [1].
>
> In the quantization setting, training corresponds to quantization-aware training (QAT). Since QAT for diffusion models also can utilize the same timestep-wise reconstruction loss used in post-training quantization, it is reasonable to expect that gradient conflict may similarly arise during QAT. Investigating the extent of this phenomenon and adapting our method to QAT is a promising direction. However, this lies beyond the scope of the current work, as QAT for diffusion models is substantially more computationally expensive than post-training quantization setting that we focus in this paper.
>
> ---
>
> > **Q2** : While the authors ablate the results on the validation set size, I hope the authors to give some insights on the whole quantization dataset's size: when the whole dataset's size goes up, there might be more weights to learn, which might be a burden. I hope the authors can clarify on this.?
>
>
> **Answer:**
> In post-training quantization, it is typically assumed that only a relatively small calibration dataset is available. While the overhead of learning sample weights increases with the calibration size, it remains relatively modest compared to the overall quantization cost. This is because quantizing diffusion models with more calibration data inherently requires a longer optimization process to reach convergence, which dominates the overall cost.
>
> Empirically, we also observe diminishing returns when increasing the calibration-set size for the 4-bit and 8-bit ImageNet 256×256 setting, as shown below for ImageNet dataset setting 4/8.  When substantially larger datasets are available, quantization-aware training (QAT) becomes a more suitable alternative; however, QAT is significantly more computationally demanding than PTQ. In such scenarios, the additional overhead introduced by our weighting strategy becomes negligible relative to the inherent cost of QAT.
>
> Finally, we emphasize that quantization is only a **one-time procedure**, and PTQ is already highly efficient.
>
> ---
>
>
> | **Data Size** | **W/A** | **FID ↓** | **sFID ↓** |
> |----------|---------|------------|-------------|
> | 2.5k     | 4/8     | 10.01      | 7.25        |
> | 5k       | 4/8     | 9.96       | 7.55        |
>
> ---
>
> ## **References**
>
> [1] Ma, Q., Ning, X., Liu, D., Niu, L., and Zhang, L. *Decouple-Then-Merge: Finetune Diffusion Models as Multi-Task Learning.* CVPR 2025, pp. 23281–23291.

---

> > ### Comment · Reviewer_DoWQ · 2025-11-24
> >
> > I sincerely thank the authors for the point-to-point rebuttals, which solves my questions. Therefore, I remain accept.

---

> > > ### Author Response · Authors · 2025-11-24
> > > **Response to Reviewer DoWQ**
> > >
> > > Dear Reviewer DoWQ,
> > >
> > > We thank the Reviewer for the valuable feedback. We greatly appreciate the time and effort the Reviewer dedicated to considering our paper and our response.

---

### Official Review · Reviewer_bPXm · 2025-11-01

**Soundness:** 3
**Presentation:** 2
**Contribution:** 3
**Rating:** 6
**Confidence:** 2

**Summary:**

Many existing post-training quantization (PTQ) methods for diffusion models assume calibration data should be treated uniformly across timesteps. This paper shows that, during PTQ, the quantization-loss gradients do not align across timesteps; treating timesteps equally can therefore cause degrade performance. To address this, the authors assign a learnable weight to each calibration sample to reflect its contribution to the gradient update, and they optimize these weights as a proxy objective to align gradient directions across timesteps.

**Strengths:**

- Improvement of FID and sFID is verified by experiment.
- Provides a theoretical justification for the proxy objective with approximation, and reports optimization trends consistent with the theory.

**Weaknesses:**

- Limited preliminaries on the specific techniques used (e.g., AdaRound) and related design choices.
- Evaluation metrics lean heavily on FID, leaving diversity aspects less explored in the main tables.

**Questions:**

- In Figure 2, is the x-axis ordered by sample timesteps? If so, it’s hard to strictly verify the claim that samples with stronger gradient alignment receive higher emphasis; a different visualization might make this clearer.
- FID and sFID indicate improved fidelity. However, because the method reweights contributions per calibration sample, I’m concerned about possible effects on sample diversity. Table 6 reports Precision; could you also report Recall?
- Similarly, for the main results, consider adding diversity-aware metrics (e.g., Precision/Recall curves) alongside FID/sFID.

---

> ### Author Response · Authors · 2025-11-21
> **Response to Reviewer bPXm**
>
> We greatly appreciate the time and effort the Reviewer dedicated to considering our paper. Here are our responses to all concerns raised by the Reviewer.
>
> ---
>
> > **W1** : Limited preliminaries on the specific techniques used (e.g., AdaRound) and related design choices.
>
> **Answer:**
> AdaRound [1] is the standard quantization scheme adopted in prior diffusion PTQ frameworks [2,3]. Therefore, to ensure a fair and consistent comparison, we follow the same quantization pipeline. We have clarified this design choice in the revised manuscript (lines 314–319).
>
> ---
>
> > **W2**: Evaluation metrics lean heavily on FID, leaving diversity aspects less explored in the main tables.
>
> **Answer:**
> We thank the reviewer for the valuable feedback. In the revised manuscript, we include additional evaluations using Precision and Recall (Table 2 and Table 6).
>
> ---
>
> > **Q1** : In Figure 2, is the x-axis ordered by sample timesteps? If so, it’s hard to strictly verify the claim that samples with stronger gradient alignment receive higher emphasis; a different visualization might make this clearer.
>
> **Answer:**
> To improve interpretability, we redesigned Figure 2 by sorting all samples in descending order of their optimized weights. We then uniformly partitioned the samples into 50 groups and plotted two curves:
> 1. the gradient alignment score of each group, computed as the average gradient cosine similarity between samples in the group and the validation samples, and
> 2. the average sample weight of the group.
>
> This revised visualization clearly illustrates the positive correlation between gradient alignment and the learned sample weights.
>
> ---
>
> > **Q2** : FID and sFID indicate improved fidelity. However, because the method reweights contributions per calibration sample, I’m concerned about possible effects on sample diversity. Table 6 reports Precision; could you also report Recall?
>
> **Answer:**
> Please see W2 above.
>
> ---
>
> > **Q3** : Similarly, for the main results, consider adding diversity-aware metrics (e.g., Precision/Recall curves) alongside FID/sFID.
>
> **Answer:**
> Please see W2 above.
>
> ---
>
> ## **References**
>
> [1] Markus Nagel, Rana Ali Amjad, Mart Van Baalen, Christos Louizos, and Tijmen Blankevoort. *Up or down? Adaptive rounding for post-training quantization.* ICML 2020.
>
> [2] Huang, Yushi, et al. *TFMQ-DM: Temporal feature maintenance quantization for diffusion models.* CVPR 2024.
>
> [3] Yefei He, Luping Liu, Jing Liu, Weijia Wu, Hong Zhou, and Bohan Zhuang. *PTQD: Accurate post-training quantization for diffusion models.* NeurIPS 2023.

---

### Official Review · Reviewer_Qghd · 2025-11-02

**Soundness:** 2
**Presentation:** 2
**Contribution:** 2
**Rating:** 4
**Confidence:** 4

**Summary:**

The paper proposes a novel post-training quantization (PTQ) method for diffusion models to improve inference speed and reduce memory consumption without retraining. Existing PTQ approaches treat all timesteps and calibration samples equally, which leads to suboptimal results because different timesteps contribute unequally and require distinct gradient directions during quantization. To address this, the authors introduce a timestep-aware weighting strategy that learns to assign optimal weights to calibration samples, aligning gradients across timesteps for better quantization. Experiments on CIFAR-10, LSUN-Bedrooms, and ImageNet demonstrate that this approach outperforms previous PTQ methods for diffusion models.

**Strengths:**

- The paper identifies a key limitation in existing PTQ methods: uniform treatment of all timesteps. It then proposes a principled weighting mechanism that learns optimal calibration weights, effectively aligning gradients across timesteps.
- The proposed method is evaluated on multiple benchmark datasets (CIFAR-10, LSUN-Bedrooms, and ImageNet), consistently outperforming prior PTQ approaches.

**Weaknesses:**

- As of 2025, most state-of-the-art diffusion models are built upon the DiT architecture. However, this submission does not include experiments on such models, which limits the generalizability and relevance of the findings to current diffusion frameworks.

- The experimental evaluation is primarily conducted on small-scale datasets (e.g., CIFAR) with low-resolution images (e.g., 32×32). While these settings are useful for preliminary validation, they do not sufficiently demonstrate the scalability or robustness of the proposed method on more challenging benchmarks.

- The study focuses mainly on bit-width configurations (e.g., W4A8, W4A32), but the results do not show clear improvements over existing baselines such as TFMQ-DM (2024). A more comprehensive comparison and discussion of potential advantages (e.g., efficiency, training stability, lower bit-width, or qualitative sample quality) would strengthen the contribution.

**Questions:**

Please refer to the weaknesses.

---

> ### Author Response · Authors · 2025-11-21
> **Response to Reviewer Qghd**
>
> Dear Reviewer Qghd,
>
> We greatly appreciate the time and effort the Reviewer dedicated to considering our paper. Here are our responses to all concerns raised by the Reviewer.
>
> ---
>
> > **W1**: As of 2025, most state-of-the-art diffusion models are built upon the DiT architecture. However, this submission does not include experiments on such models, which limits the generalizability and relevance of the findings to current diffusion frameworks.
>
> **Answer**: Our primary experiments follow the standard evaluation protocol adopted in prior diffusion PTQ works such as TFMQ-DM [1] and PTQD [2]. To further assess generalizability to larger architectures, we additionally integrate our method into a DiT-based diffusion model (PTQ4DiT [3]). Our method improves upon PTQ4DiT, demonstrating that it provides complementary benefits when applied to transformer-based diffusion architectures.
>
> | **Settings** | **W/A** | **FID ↓** | **sFID ↓** |
> |-------------|---------|-----------|------------|
> | PTQ4DiT [3]     | 4/8     | 9.17      | 24.29      |
> | **Ours**    | 4/8     | **8.65**  | **23.84**  |
>
> ---
>
> > **W2**: The experimental evaluation is primarily conducted on small-scale datasets (e.g., CIFAR) with low-resolution images (e.g., 32×32). While these settings are useful for preliminary validation, they do not sufficiently demonstrate the scalability or robustness of the proposed method on more challenging benchmarks.
>
> **Answer**: We thank the reviewer for the valuable feedback. Beyond CIFAR-10, our original submission already includes experiments on LSUN-Bedrooms (256×256) and ImageNet (256×256), which are standard high-resolution benchmarks for diffusion model quantization. To further strengthen the evaluation, we additionally report results on the FFHQ 256×256 dataset in table below. Across all three higher-resolution datasets, our method consistently improves quantization performance, demonstrating strong scalability and robustness.
>
> **Table X: Quantization results for image generation with LDM-4 on FFHQ at resolution 256×256.**
>
> | Methods      | Bits (W/A) | FID ↓ | sFID ↓ |
> |--------------|------------|--------|---------|
> | Full Prec.   | 32/32      | 9.36   | 8.67    |
> | PTQD [2]     | 4/32       | 12.01  | 11.12   |
> | TFMQ-DM [1]  | 4/32       | 9.55   | 9.70    |
> | **Ours**     | 4/32       | **9.14** | **9.58** |
> | PTQD [2]     | 4/8        | 11.42  | 11.43   |
> | TFMQ-DM [1]  | 4/8        | 9.65   | 9.74    |
> | **Ours**     | 4/8        | **9.21** | **9.63** |
>
>
>
> ---
>
> > **W3**: The study focuses mainly on bit-width configurations, but the results do not show clear improvements over existing baselines such as TFMQ-DM (2024). A more comprehensive comparison and discussion of potential advantages (e.g., efficiency, training stability, lower bit-width, or qualitative sample quality) would strengthen the contribution.
>
> **Answer**:
> We thank the reviewer for the valuable feedback. In the revised manuscript, we provide a more comprehensive comparison in Tables 2 with additional qualitative metrics. Our method achieves consistent improvements across benchmarks. For low-bit configurations, Table 6 of the revised paper shows that our approach substantially outperforms existing baselines. We have also clarified the computational overhead in lines 466–476, where our method is shown to be slightly slower than TFMQ-DM [1] but faster than Q-Diffusion [4], offering a favorable efficiency–performance trade-off.
>
> ---
>
> **Reference**
>
> [1] Huang, Yushi, et al. *TFMQ-DM: Temporal feature maintenance quantization for diffusion models.* CVPR 2024.
>
> [2] Yefei He, Luping Liu, Jing Liu, Weijia Wu, Hong Zhou, and Bohan Zhuang. *PTQD: Accurate post-training quantization for diffusion models.* NeurIPS 2023.
>
> [3] Junyi Wu, Haoxuan Wang, Yuzhang Shang, Mubarak Shah, and Yan Yan. *PTQ4DiT: Post-training quantization for diffusion transformers.* NeurIPS 2024.
>
> [4] Xiuyu Li, Yijiang Liu, Long Lian, Huanrui Yang, Zhen Dong, Daniel Kang, Shanghang Zhang, and
> Kurt Keutzer. Q-Diffusion: Quantizing diffusion models. ICCV 2023

---

### Author Response · Authors · 2025-12-04
**Summarize the discussion**

Dear Area Chair,

Thank you for taking the time to review the discussion during the rebuttal period. Below is a concise summary of the reviewers’ engagement and how their concerns were addressed.

---

### **Summary of the Rebuttal Discussion**

We sincerely thank the Reviewers for their thoughtful and constructive feedback. We are encouraged that the Reviewers acknowledge several strengths  of our work:

- Reviewer Qghd: Recognized that the paper proposes a principled method addressing an important limitation in existing PTQ approaches.
- Reviewer bPXm: Appreciated the  theoretical justification, the consistency between theory and observed optimization behavior, and the empirical improvements.
- Reviewer DoWQ: Highlighted the strong and meaningful motivation, the clarity and completeness of the method’s design and theoretical support, and the significant performance gains validating the approach.

Only Reviewer DoWQ actively participated in the discussion before the system outage and remained fully satisfied, keeping their **score of 8**.

Reviewer bPXm, although not active during the rebuttal period, had already provided a positive score of 6, and we addressed their main concern about image diversity by adding Recall metrics.

Reviewer Qghd, who assigned a score of 4, did not participate in the discussion. Their requests mainly concerned broader evaluation, and we provided substantial additional results that directly addressed these points.

The revised paper also corrects citations, revised visualization, and includes new experiments, all highlighted in light blue in the updated PDF.

---

### **Detailed Summary**
#### **1. Reviewer Qghd (score 4)**
Reviewer Qghd asked for evidence that our method generalizes to diffusion-transformer (DiT) architectures. In response, we integrated our approach into the PTQ4DiT pipeline and added a DiT-based result, demonstrating that our method remains effective and provides complementary improvements on a DiT diffusion model.

The reviewer also requested evaluation on higher-resolution datasets. In the rebuttal, we added an additional high-resolution dataset (FFHQ 256×256) to complement the existing ImageNet 256×256 and LSUN-Bedroom 256×256 results already included in the original submission.

Finally, the reviewer asked for more comprehensive evaluation. We provided additional qualitative and quantitative results, including Precision and Recall (Tables 2 and 6), as well as lower bit-width experiments (Table 6). Our method demonstrates improvement on most settings.

The reviewer did not follow up before the system issue, so their final stance is unknown. However, we believe our response directly and comprehensively addressed all concerns.

---

#### **2. Reviewer bPXm (score 6)**
Reviewer bPXm raised the question of whether fidelity improvements might reduce diversity. We clarified this by providing Recall scores in Tables 2 and 6, where our method improves diversity in most settings.

We also revised the visualizations as requested.
The reviewer did not follow up before the system issue, but all concerns were explicitly resolved.

---

#### **3. Reviewer DoWQ (score 8)**
Reviewer DoWQ provided a strongly positive evaluation with only minor questions (citations, visualization clarity, and possible extensions).
They were the only reviewer to engage actively in the rebuttal and expressed full satisfaction afterward, keeping their score unchanged.

---
We hope this summary is helpful in your final assessment.
Thank you again for your time and for coordinating the review process.
Sincerely,

Authors

---

### Meta-Review · Area_Chair_7h5o · 2026-01-06

**Summary:**

This paper proposes a novel post-training quantization (PTQ) method for diffusion models that addresses timestep-wise heterogeneity by learning adaptive weights for calibration samples. The reviewers acknowledged the importance of the problem, the consistency between theoretical justification and empirical behavior, and the strong performance gains across standard benchmarks. The rebuttal provided additional experiments, corrected presentation issues, and expanded evaluations to address several reviewer concerns. Overall, the majority of reviewers expressed confidence in the technical soundness and effectiveness of the approach, while the concerns about the evaluation were addressed by including additional baselines and datasets.

**Reviewer Concerns:**

Reviewer Qghd questioned whether the proposed method generalizes to diffusion-transformer (DiT) architectures and higher-resolution datasets. In response, the authors added DiT-based experiments within the PTQ4DiT pipeline, as well as additional high-resolution results (i.e., FFHQ 256×256), and expanded both quantitative and qualitative evaluations with Precision, Recall, and lower bit-width settings. Overall, the authors’ rebuttal addressed the stated concerns.

Reviewer bPXm raised a potential trade-off between fidelity improvements and sample diversity. The authors responded by adding Recall metrics and revising visualizations, showing that diversity is preserved or improved in most settings. Reviewer DoWQ provided a strongly positive assessment, highlighting the clarity of the method design, the strength of the theoretical support, and the significance of the empirical gains, and remained fully satisfied after the rebuttal.

**Reviewer Scores:**

The authors addressed the concerns raised by reviewer Qghd (Score: 4) by conducting evaluations with additional baselines and datasets, while the other two reviewers were very positive on this paper.

---

### Decision · Program_Chairs · 2026-01-26

Accept (Poster)